# A scoping review protocol to map the evidence on interventions to prevent overweight and obesity in children

Peer-Benedikt Vincent Bussiek, Chiara De Poli, Gwyn Bevan

Department of Management, London School of Economics and Political Science, London, UK

**Correspondence to**
Chiara De Poli;
c.de-poli@lse.ac.uk

## ABSTRACT

**Introduction** Obesity has become one of the biggest public health problems of the 21st century. Prevalence of obesity in children and adolescents has increased dramatically worldwide over the last 20 years, and this trend is expected to continue. Obesity in childhood is concerning as it predicts obesity in adulthood, a common risk factor for a wide array of chronic diseases and poor health outcomes. Obesity is preventable and a vast but fragmented body of evidence on preventative interventions is now available. This article outlines the protocol for a scoping review of published literature reviews on interventions to prevent obesity in children. The scoping review addresses the broad research question 'What is the evidence on interventions to prevent childhood obesity?'. It aims to give an overview of the various interventions available, understand those which are effective and identify barriers and facilitators to their effectiveness.

**Methods and analysis** The six-staged Arksey and O'Malley methodology framework is used to guide the scoping review process: following the definition of the research questions (stage 1); the eligibility criteria and search strategy are defined (stage 2); the study selection process based on the eligibility criteria identified will follow (stage 3); a framework developed for this review will then inform the extraction and charting of data from the included reviews (stage 4); results will be aggregated and summarised with criteria relevant for health professionals and policy-makers (stage 5); and the optional consultation (stage 6) exercise is not planned.

**Ethics and dissemination** Since the scoping review methodology aims at synthetising information from available publications, this study does not require ethical approval. An article reporting the results of the scoping review will be submitted for publication to a scientific journal, presented at relevant conferences and disseminated as part of future workshops with professionals involved in obesity prevention.

## BACKGROUND

Childhood obesity, the abnormal or excessive fat accumulation that may impair health,[1] has become a global problem. In 2015, about 110 million children and young adults (under 20 years of age) were estimated to be obese, equivalent to an overall prevalence of 5%.[2] Epidemiological data show that the number of overweight or obese infants and

young children (aged 0–5 years) increased from 32 million globally in 1990 to 42 million in 2013. If these trends will continue, the number of overweight or obese infants and young children is expected to spiral up to 70 million by 2025.[1]

### Strengths and limitations of this study

► The present protocol is for a scoping review of published systematic reviews, as a pragmatic way of dealing with the large evidence base available on obesity prevention in children.
► The present scoping review aims to use 11 electronic databases and tailored search strings which have been iteratively refined in order to increase the likelihood of retrieving as many relevant published article reviews as possible.
► Only article reviews published in English in scientific journals will be considered eligible for inclusion.
► A quality assessment of the article reviews included in the scoping review will not be performed, as this would be beyond the aim of a scoping review.

Although the prevalence of childhood obesity is estimated to be lower than the prevalence of adult obesity (5% against 13%), the rate of increase in childhood obesity in many countries is alarmingly greater than the rate of increase in adult obesity.[2] These trends are expected to continue if no radical actions to tackle the epidemic are implemented.

Obesity in childhood can affect a child's immediate health, as it is associated with a number of physical and psychological comorbidities (eg, asthma, dental caries, attention-deficit hyperactivity disorder and non-alcoholic fatty liver disease).[3–6] It can also impair educational attainment[7–9] and quality of life[10 11] and can have substantial long-term health consequences into adulthood.[12] There is evidence that an early adiposity rebound, the point in life when body mass index (BMI) rises again after reaching a nadir, predicts later obesity.[13–15] Hence, obese children are more likely than children with normal BMI

to remain obese as adults[12 16–20] and to experience greater risk of poorer health outcomes.[21 22]

Options for treating obesity, pharmacological and surgical, are currently available but are costly, hence cannot be afforded at scale, and can have complications. For these reasons, early prevention during childhood is better than attempts at cure later in life.[23]

However, obesity prevention in children is a complex task. The obesity system map for the UK represents such complexity in a powerful way. The map was developed as a heuristic tool to describe the anatomy of the obesity system and it includes more than 100 variables (clustered thematically around physiology, individual physical activity, the physical activity environment, food consumption, food production, individual psychology and social psychology) and several intricate loops representing causal linkages between the different variables.[24] To confront this complexity, strategies to tackle childhood obesity that are currently implemented in many countries combine behavioural, lifestyle interventions targeting particular subpopulations and initiatives addressing the obesogenic environment to which the wider population is exposed.[25] As such they intend to address the energy imbalance that leads to obesity, caused by a combination of exposure to an unhealthy environment as well as inadequate behavioural responses to that environment and the specific needs of a person.[26–28]

For example, in England, lifestyle weight management services are available for overweight or obese children and young people[29] alongside population-based interventions, such as a soft drink industry levy, improved food labelling and a reduction of sugar in the products children eat most, which have been pledged in the plan for action against childhood obesity published by the UK Government in 2016.[30] Similar strategies are implemented in the USA,[31] Canada[32] and in many European countries.[33]

In order to inform evidence-based policies in the area of obesity prevention in children, a synthesis of the body of evidence currently available is needed.[13 34] To this aim, a scoping review of reviews of the literature on interventions to prevent childhood obesity is proposed with the objective of providing a descriptive overview of what these interventions are, if they are effective and why they succeed (or do not).

The scoping review represents an appropriate methodology for reviewing large bodies of literature in order to generate an overview of research undertaken on a topic and determine the range of studies that are available, summarise research results and identify evidence gaps.[35] As such, they do not aim at critically appraising individual studies which may be in fact heterogenous in terms of study design, methodology and, hence, quality of the results reported.[36] Despite this limitation, a scoping review of the literature on the topic identified will be valuable for at least two reasons. First, the present scoping review aims to overcome the narrow foci of the few scoping reviews already available on prevention of childhood obesity (eg, interventions delivered in specific settings, such as schools)[37–40] and to adopt a comprehensive approach to the topic. Second, over the last 10 years, several systematic reviews have been published,[41–44] and a synthesis of this growing evidence base is now due. In this article, the protocol that will inform the scoping review is presented.

## METHODS AND ANALYSIS
### Protocol design
The scoping review is informed by the framework proposed by Arksey and O'Malley[36] which has been further developed by Levac et al[45] and the Joanna Briggs Institute.[46] This recommends to organise the review process in at least five stages.[36]

► Stage 1. Identifying the research question.
► Stage 2. Identifying relevant studies.
► Stage 3. Study selection.
► Stage 4. Charting the data.
► Stage 5. Collating, summarising and reporting the results.

The original framework proposed by Arksey and O'Malley suggests an optional consultation exercise (stage 6) with key stakeholders in order to identify additional references about potential studies to include as well as collect their feedback about the findings uncovered by the scoping review. Although a consultation with key stakeholders would represent a valuable exercise, the present scoping review will not encompass one because of time and budget constraints.

### Stage 1: identifying the research questions
Preliminarily to identifying the research question, an exploratory review of the literature on childhood obesity helped refine the scope of the present protocol. This phase informed the decision not to use any criteria to restrict the review to specific study populations (eg, specific age groups) as it became clear that overweight and obesity need to be addressed as early as possible and also opportunistically, as weight gain in children appears to be a good predictor of obesity in adulthood.[12 16–20]

On the basis of the initial exploratory research, the following research questions were identified:

1. What types of interventions to prevent children's obesity are addressed in the literature?
2. What are the children populations targeted by these interventions?
3. In what settings are these interventions provided?
4. Are these interventions effective?
5. Which measures are used to assess obesity in children?
6. If these interventions are effective, what is the scale of the reduction in childhood obesity?
7. What are the barriers and facilitators to effective implementation of these interventions?
8. What evidence is there of the effectiveness of these interventions when they are combined?

## Stage 2: identifying relevant studies

Following the framework of Arksey and O'Malley, the second stage of the scoping review process aimed to identify the criteria that will be used to select the studies for inclusion in the review. Although a scoping review is designed to cover a broad spectrum of literature, these criteria will guide the search and help filter for relevant sources.

The scoping review will include published systematic reviews that can be retrieved from the following electronic databases: Cochrane Database of Systematic Reviews, Cumulative Index to Nursing and Allied Health Literature, Education Resources Information Center, Google Scholar, Joanna Briggs Library, MEDLINE/PubMed, National Health Service Economic Evaluation Database, PsycInfo, Scopus, SocIndex and Web of Science. Reference lists of reviews found through the electronic search will be checked to ensure that relevant articles are included in the scoping review.

Based on the initial exploratory research, we agreed the following eligibility criteria:
► Type of publication: journal articles.
► Time frame: any.
► Language: English.
► Study population: children and adolescents, aged below 18.
► Types of intervention: interventions aiming at preventing childhood obesity.
► Types of review articles: systematic reviews, meta-analyses, scoping reviews, evidence maps, rapid reviews, literature reviews, evidence syntheses, reviews of reviews, narrative reviews and critical reviews.

Based on the initial scoping process, it was agreed to exclude: conference abstracts, book reviews, commentaries or editorial articles, reviews focusing on the adult population (individuals aged 18 year or above) and reviews focusing on interventions for the treatment of obesity (eg, bariatric surgery), rather than its prevention.

As suggested by Levac et al,[45] the team used an iterative process to identify also key search terms. Initially, the following keywords were used: child*, obes*, weight*, intervention, prevent* and review. The review articles retrieved were then screened for their titles, abstracts and index terms. An academic librarian was consulted and advised on the most appropriate Medical Subject Headings terms for the search and how to modify them for the different databases used. Based on this exploratory scoping phase, the search strings for each database were finalised (online supplementary material 1). Articles were retrieved from each database and imported into a reference management software.

## Stage 3: study selection

The third stage of the framework of Arksey and O'Malley's framework aims to identify the studies that will be included in the scoping review. The team consolidated the results of the searches run on the different databases and removed studies retrieved from more than one database

in order to exclude duplicates. A member of the team will then screen titles and abstracts of the articles to exclude those that do not meet the eligibility criteria identified in the second stage of the protocol. For those fulfilling the eligibility criteria, the full article will be retrieved.

A sample (ie, 20%) of the retrieved articles will be screened by another team member to ensure a consistent application of the eligibility criteria for inclusion in the review. Titles and abstracts of the articles for which the first reviewer could not determine whether they are eligible for inclusion will also be reviewed. Disagreements about study eligibility of the sampled articles will be discussed between the two reviewers until consensus is reached or by arbitration of a third reviewer, if required. The process of study selection is reported using a Preferred Reporting Items for Systematic Reviews and Meta-Analyses flow chart, which will be updated once the review is completed (online supplementary material 2).

## Stage 4: charting the data

Based on the preliminary scoping phase, a data extraction framework was developed. It includes 19 categories that will be used to assess the full review articles retrieved from the literature fulfilling the eligibility criteria for inclusion (table 1).

Alongside standard bibliographical information (ie, authors, title, journal and year of publication), type and objectives of the review will be reported. For each article, information on the interventions covered by the review, characteristics of the study populations, definition of overweight/obesity adopted in the reviews, setting, length and intensity of the interventions, types of outcomes assessed, information on the effectiveness of the interventions and facilitators and barriers for the implementation of the interventions will be tabled.

The framework will be pilot tested by two team members on a sample of the included studies (ie, 10% of the complete list of retrieved studies) in order to ensure that the coding framework is consistently applied. If necessary, the categories will be modified and the data extraction framework revised accordingly. Questions arising when piloting the framework will be discussed by the team and possible disagreement will be resolved through team consultations.

The same two members of the team will be in charge of independently charting the data from each included review study, following the data extraction framework. In order to ensure inter-rater reliability, a sample (ie, 20%) of the included articles independently reviewed will then be compared by the two members of the team. Discrepancies in extracted data will be discussed between the two reviewers until consensus is reached or by arbitration of a third reviewer, if required.

## Stage 5: collating, summarising and reporting the results

The analysis of the data collected using the data extraction framework will provide information on the body of research that has been conducted on interventions

**Table 1** Data extraction framework

| Main category | Subcategory | Description |
|---|---|---|
| 1. Authors | | |
| 2. Title | | |
| 3. Journal | | |
| 4. Year of publication | | |
| 5. Objective(s) of the review | | Describe the stated objectives of the review |
| 6. Type of review | | Specify eg, if systematic review, meta-analysis, scoping review, narrative review or other |
| 7. Number of studies included in the review | | Indicate the number of primary studies included in the review |
| 8. Years of publication of the studies included in the review | | Specify the range of the years of publications of the studies included in the review |
| 9. Countries where the studies included in the review were conducted | | Specify the geographical areas covered by the studies included in the review |
| 10. Type of studies included in the review | | Specify if the review includes specific types of studies (eg, RCTs, cost-effectiveness analyses, qualitative studies, modelling studies) |
| 11. Description of the intervention(s) | Type of intervention | Specify the type(s) of the interventions on which the review focuses (eg, lifestyle interventions, weight management programmes and sugar taxes) |
| | Delivery of the intervention | Describe how and by whom the intervention is delivered |
| | Length and intensity of the intervention | Describe for how long the intervention is delivered and its intensity |
| 12. Definition of overweight/obesity | | Specify the definition of overweight/obesity used in the review |
| 13. Description of the study population | Target population | Specify if the intervention (i) targets individuals within subpopulation groups or (ii) the broad population (eg, in case of interventions on the built environment, national policies and regulation) |
| | By BMI (preintervention) | Specify the distribution of the study population by BMI preintervention |
| | By age | Specify the age groups covered by the review |
| | By sex | Specify if the review focuses on interventions targeting specifically boys or girls, or indicate the distribution of study participants by sex |
| | By ethnic background | Specify if the review focuses on interventions targeting specific ethnic groups (eg, South Asians), or indicate the distribution of study participants by ethnic groups |
| | By socioeconomic background | Specify if the review focuses on interventions targeting children living in deprived areas, or indicate the distribution of study participants by socioeconomic background |
| | By other characteristics of the study population (eg, disability, comorbidity) | Specify if the review focuses on specific populations (eg, children with disabilities) |
| 14. Setting of the intervention(s) | | Specify if the review focuses on interventions delivered in school-based, family-based or community-based settings |
| 15. Reported outcomes | | Describe the intervention outcomes reported in the review (eg, weight, BMI, self-efficacy) |
| 16. Effectiveness | | Describe the results reported in the review (eg, change in BMI) |

Continued

**Table 1** Continued

| Main category | Subcategory | Description |
| --- | --- | --- |
| 17. Impact | | Describe the distribution of the study population by BMI (or other relevant outcomes) postintervention |
| 18. Facilitators | | Describe the factors that support or enable the implementation of the intervention reported in the review |
| 19. Barriers | | Describe the factors that inhibit the implementation of the intervention reported in the review |

BMI, body mass index; RCTs, randomised controlled trials.

to prevent obesity in children. For example, evidence on the interventions to tackle children obesity will be presented by age groups in order to show at what point in the life course and what types of interventions are more effective and, hence, worth pursuing. Also, it will be possible to highlight the clinical effectiveness of the interventions reviewed (eg, the change in the BMI of participants included in the review studies) and their material impact on the population in need (eg, how the distribution of study participants by BMI groups changes before and after the intervention). Conversely, it will also show areas that have been under researched and may require further investigation. Results will be presented in an aggregate and visual form (eg, using tables and charts), as appropriate.

## ETHICS AND DISSEMINATION

Since the scoping review methodology aims at synthesising information from publicly available publications, this study does not require ethical approval. In terms of dissemination activities, an article reporting the results of the scoping review will be submitted for publication to a scientific journal and presented at relevant conferences. We anticipate the results of the scoping review to provide a comprehensive overview of the evidence base of interventions to prevent obesity in children and to highlight areas where evidence is controversial or missing. It will also provide key information to policy makers and health professionals interested in planning, funding and delivering evidence based and effective interventions to prevent children obesity. For this reason, the results will be also disseminated as part of future workshops with professionals involved in obesity prevention.

**Contributors** P-BVB contributed to develop the research questions and the methods and contributed substantially to the drafting and editing. CDP conceived of the idea the scoping review, developed the research questions and contributed to the development of the methods. She contributed extensively to the drafting and editing of the manuscript. GB supervised the preparation of the protocol and critically reviewed the manuscript. All authors have approved the final manuscript.

**Funding** P-BVB and GB were funded by the Department of Management at the London School of Economics and Political Science. CDP was funded by the National Institute for Health Research Collaboration for Leadership in Applied Health Research and Care (NIHR CLAHRC) North Thames at Barts Health NHS Trust.

**Disclaimer** The views expressed are those of the authors and not necessarily those of the NHS, the NIHR or the Department of Health.

**Competing interests** None declared.

**Patient consent** Not required.

**Provenance and peer review** Not commissioned; externally peer reviewed.

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
