## [Reviewer comments · BMJ Open]

ARTICLE DETAILS

TITLE (PROVISIONAL)	A scoping review protocol to map the evidence on interventions to prevent overweight and obesity in children
AUTHORS	Bussiek , Peer-Benedikt Vincent De Poli, Chiara Bevan, Gwyn

VERSION 1 – REVIEW

REVIEWER	Dr Amanda Avery University of Nottingham UK
REVIEW RETURNED	23-Oct-2017

GENERAL COMMENTS	dear authors this proposed scoping review will involve much work! general comments; please avoid the use of the first person in written expression for a scientific publication. For example in the abstract - 'The five-staged A&) framework is used to guide.....Following the definition of the research questions, the eligibility criteria and search strategy will be defined.It is anticipated that the results of the scoping review will....' etc (there are many places throughout the script where the first person is used inappropriately. also do not refer to he/she - instead use 'they'. Also a general comment; my understanding is that with respect to 'tracking of obesity' It is more obesity in adolescence which tracks through to obesity in adulthood - or that is why the current evidence lies. Yes indeed the NCMP data (ref 24) does show an increasing prevalence of obesity from the age of 4 to 11 but this is relatively new data. we need to be respectful, and balanced, that not all children who are overweight or who present with obesity will become adults with obesity. There were a number of places where I felt that 'children' needed to be replaced by 'adolescents' when you were referring to tracking of obesity into adulthood. With respect to the research aim/objectives, these could be better reflected in the abstract - so the last sentence of the abstract introduction: 'this scoping review aims to and then a summary of lines 36-44 in the main body. In terms of limitations - yes the scoping review provides a pragmatic methodology for dealing with a wealth of information but I do feel that you need to be more honest about the fact that the methodology may well overlook some of the detail of the original studies, including the quality and thus robustness of the data and this may influence
---

	interpretation? line 55 - include NAFLD line 28/9 - hence cannot BE afforded....
--	---

REVIEWER	Elise C. Brown School of Health Sciences, Oakland University, USA
REVIEW RETURNED	03-Dec-2017

GENERAL COMMENTS	This is an interesting protocol that seeks to conduct a scoping review of child obesity interventions. As there have been many reviews that have investigated the effectiveness of child obesity interventions, this study aims to synthesize the findings of these reviews. Overall, there are a number of points for the author to address as mentioned below. One particular issue is the disjointedness of the “Background” section as it does not flow very well. Abstract Page 3, lines 6 – 21: The abstract introduction was informative and the rationale was developed, however, the research question was not provided. Can you please clearly state the research question the scoping review aims to answer? Page 3, lines 24 - 38: For the “Methods and Analysis” section, did you consider using the optional 6th stage of Arksey and O'Malley's framework? Page 3, lines 40 - 49: For the “Ethics and Dissemination” section, there was no mention of ethical approval of the study nor a plan of how the results will be disseminated. Can you please comment on these points? Background Page 5, lines 6 – 13: The background opens with 2 stand-alone sentences and then a paragraph follows. This should be revised so that the stand-alone sentences are combined to a single paragraph or combined with the following paragraph. Page 5, lines 11 – 13: The sentence, “Most of the world's population live in countries where overweight and obesity kill more people than underweight,” is a direct quote from the cited WHO website, but direct quotes were not used. Please paraphrase. Page 5, lines 34 – 36: The sentence mentioning educational attainment and quality of life does not appear to align the reference provided. Can you please clarify? Page 5, lines 41 – 48: This paragraph seems out of place. It discusses epidemiological data and should be grouped with the first paragraph in the introduction that has related information. Page 6, lines 22 – 26: For the sentence arguing that overweight and obese children and teens are more likely to become obese adults, can you please provide a reference? Page 6, line 28: The word “nowadays” is too informal, please revise. Page 6, lines 36 – 38: You mentioned the cause of obesity relating to an energy imbalance in the beginning of the background, and now you are discussing it again and mentioning other factors. Please organize so that the background logically flows better and also provide a reference for this suggested etiology. Page 6, lines 38 – 47: This sentence is confusing. The way that it is currently written, I do not think that the inclusion of “e.g.” is necessary, and it is unclear what “as well as critical developmental,
--

	biological or behavioural factors over the life-course of individuals” is specifically referring to. Please revise. Page 7, line 10: Should this read UK government? Please be specific. One thing that is missing from the background is what other similar reviews have found such as the review conducted by Khambalia et al. (2012) and how the proposed review is different and therefore needed. Methods Page 8, lines 8 – 25: Include a sub-heading under “Methods” for this section. Perhaps “Protocol” or “Protocol Design”? Page 8, lines 47 – 57 and page 9, lines 3 – 21: Another question that you should consider is what obesity measure is being used as a measure of effectiveness (BMI, BMI-SDS, waist circumference, waist-to-height ratio, etc.) Page 10, line 33: Should “obes” be “obese” or “obesity”? Pages 12 – 13: Consider length of intervention as a main category or sub-category. Conclusion Rather than a conclusion, there should be an “Ethics and Dissemination” section per author instructions.
--	--

VERSION 1 – AUTHOR RESPONSE

Dear Dr Avery and Dr Brown

Thank you for taking the time to review our manuscript.

We have read carefully your recommendations and we have revised the article as suggested. In this document we discuss how we have addressed each of the comments we received. We hope you will find our responses comprehensive.

Thank you for the opportunity to revise our manuscript.

Sincerely yours,

Ben Bussiek

Chiara De Poli

Gwyn Bevan

Reviewer: 1

Reviewer Name: Dr Amanda Avery

Institution and Country: University of Nottingham, UK

Competing Interests: none

General comments

• please avoid the use of the first person in written expression for a scientific publication. For example in the abstract- 'The five-staged A&D framework is used to guide.....Following the definition of the research questions, the eligibility criteria and search strategy will be defined.It is anticipated that the results of the scoping review will....' etc (there are many places throughout the script where the first person is used inappropriately. also do not refer to he/she - instead use 'they').

Thanks for this comment. We have now revised the manuscript using a passive voice.

- Also a general comment; my understanding is that with respect to 'tracking of obesity' It is more obesity in adolescence which tracks through to obesity in adulthood - or that is why the current evidence lies. Yes indeed the NCMP data (ref 24) does show an increasing prevalence of obesity from the age of 4 to 11 but this is relatively new data. we need to be respectful, and balanced, that not all children who are overweight or who present with obesity will become adults with obesity. There were a number of places where I felt that 'children' needed to be replaced by 'adolescents' when you were referring to tracking of obesity into adulthood.

Thanks for this comment. We do not intend generalize that every overweight or obese child will become an obese adult. In our work we intend to acknowledge a body of evidence that suggests that childhood obesity is a good predictor of adult obesity.¹⁻⁶ For example, a recent systematic review and meta-analysis (including fifteen prospective cohort studies with 200,777 participants followed up) reported that obese children and adolescents were around five times more likely to be obese in adulthood than those who were not obese, around 55% of obese children go on to be obese in adolescence, around 80% of obese adolescents will still be obese in adulthood and around 70% will be obese over age 30.⁷

Interestingly, such evidence is currently used to support the development and implementation of strategies to prevent childhood obesity, see for example the WHO Report of the Commission on Ending Childhood Obesity (2016).⁸

- With respect to the research aim/objectives, these could be better reflected in the abstract - so the last sentence of the abstract introduction: 'this scoping review aims to and then a summary of lines 36-44 in the main body.

Thanks for this comment. We revised the Introduction section in the abstract in order to better explain the objectives of the scoping review.

- In terms of limitations - yes the scoping review provides a pragmatic methodology for dealing with a wealth of information but I do feel that you need to be more honest about the fact that the methodology may well overlook some of the detail of the original studies, including the quality and thus robustness of the data and this may influence interpretation?

The scoping review is an approach to evidence synthesis which can be of particular use for topics which are complex or heterogeneous.⁹ By definition, a scoping review aims to give a descriptive overview of a broad field without critically appraising individual studies which may vary greatly in terms of study design, methodology and, consequently, quality of the results reported.¹⁰ We appreciate this may be a limit of the approach we are adopting, but equally we believe that there is scope for carrying out a scoping review on the topic we have identified for at least two reasons. Firstly, the present scoping review aims to overcome the narrow foci of the few scoping reviews already available on prevention of childhood obesity (e.g. interventions delivered in specific settings, such as schools)¹¹⁻¹⁴ and to adopt a comprehensive approach to the topic. Secondly, over the last ten years, several systematic reviews have been published (see for example 15-18), and a synthesis of this growing evidence base seems now due.

We have revised the Background section of the manuscript to acknowledge the limitations intrinsic to the scoping review approach and to explain why we believe there is scope for one on the topic identified.

- line 55 - include NAFLD

Thanks for suggesting to include also NAFLD. We have now included this comorbidity, which is now also appropriately referenced (see 19,20).

- line 28/9 - hence cannot BE afforded....

We have now corrected the sentence as suggested. Thanks for spotting this.

Reviewer: 2

Reviewer Name: Elise C. Brown

Institution and Country: School of Health Sciences, Oakland University, USA

Competing Interests: None declared

This is an interesting protocol that seeks to conduct a scoping review of child obesity interventions. As there have been many reviews that have investigated the effectiveness of child obesity interventions, this study aims to synthesize the findings of these reviews. Overall, there are a number of points for the author to address as mentioned below. One particular issue is the disjointedness of the "Background" section as it does not flow very well.

• Abstract

Page 3, lines 6 – 21: The abstract introduction was informative and the rationale was developed, however, the research question was not provided. Can you please clearly state the research question the scoping review aims to answer?

Thanks for this comment. We have now revised the Introduction of the abstract which now includes also the broad research question addressed by the scoping review. In the section on the Methods of the manuscript, the research question is unpacked in 8 specific questions.

Page 3, lines 24 - 38: For the "Methods and Analysis" section, did you consider using the optional 6th stage of Arksey and O'Malley's framework?

Thanks for the comment. We have revised the Methods and Analysis section of the abstract to justify why we did not use the 6th stage of the Arksey and O'Malley's framework. This is further explained in the Methods section of the manuscript.

Page 3, lines 40 - 49: For the "Ethics and Dissemination" section, there was no mention of ethical approval of the study nor a plan of how the results will be disseminated. Can you please comment on these points?

Thank you for the opportunity to clarify the ethical implications and dissemination plan for our work. Given the nature of the scoping review we intend to carry out, an ethic approval is not required. In terms of dissemination, the results of the scoping review will be submitted for publication in a scientific journal and presented at relevant conferences (e.g. Public Health Science conference organized by the Lancet every year). We expect to organize future workshops with professionals involved in obesity prevention and the results of the scoping review will be shared also with them.

The "Ethics and Dissemination" section of the abstract has been revised accordingly.

• Background

Page 5, lines 6 – 13: The background opens with 2 stand-alone sentences and then a paragraph follows. This should be revised so that the stand-alone sentences are combined to a single paragraph or combined with the following paragraph.

Following this and other suggestions made by the reviewers, we have substantially revised the Background section of the manuscript in order to improve its readability and coherence. This comment has been addressed in the current version of the Background section.

Page 5, lines 11 – 13: The sentence, "Most of the world's population live in countries where overweight and obesity kill more people than underweight," is a direct quote from the cited WHO website, but direct quotes were not used. Please paraphrase.

Following the substantial revision of the Background section, this sentence has now been deleted.

Page 5, lines 34 – 36: The sentence mentioning educational attainment and quality of life does not appear to align the reference provided. Can you please clarify?

Thanks for this comment. In the first draft of the manuscript we referenced the wrong publication. To support the statement, we have now added relevant references: for educational attainment, 21–23 for quality of life, 24,25 for long term health consequences.⁶

□ Page 5, lines 41 – 48: This paragraph seems out of place. It discusses epidemiological data and should be grouped with the first paragraph in the introduction that has related information. Following this and other suggestions made by the reviewers, we have substantially revised the Background section of the manuscript in order to improve its readability and coherence. This comment has been addressed in the current version of the manuscript.

□ Page 6, lines 22 – 26: For the sentence arguing that overweight and obese children and teens are more likely to become obese adults, can you please provide a reference?

Thanks for this comment, which echoes a comment also by reviewer 1. In our work we intend to acknowledge a body of evidence that suggests childhood obesity is a good predictor of adult obesity.^{1–6} For example, a recent systematic review and meta-analysis (including fifteen prospective cohort studies with 200,777 participants followed up) reported that obese children and adolescents were around five times more likely to be obese in adulthood than those who were not obese, around 55% of obese children go on to be obese in adolescence, around 80% of obese adolescents will still be obese in adulthood and around 70% will be obese over age 30.⁷ Interestingly, such evidence is currently used to support the development and implementation of strategies to prevent childhood obesity, see for example the WHO Report of the Commission on Ending Childhood Obesity.⁸

□ Page 6, line 28: The word “nowadays” is too informal, please revise. We have now removed the word “nowadays” throughout the manuscript.

□ Page 6, lines 36 – 38: You mentioned the cause of obesity relating to an energy imbalance in the beginning of the background, and now you are discussing it again and mentioning other factors. Please organize so that the background logically flows better and also provide a reference for this suggested etiology. Following this and other suggestions made by the reviewers, we have substantially revised the Background section of the manuscript. Following this specific comment, we have provided additional references for energy imbalance, i.e. when the energy intake exceeds the energy consumption, as one of the causes of obesity, see 26–28.

□ Page 6, lines 38 – 47: This sentence is confusing. The way that it is currently written, I do not think that the inclusion of “e.g.” is necessary, and it is unclear what “as well as critical developmental, biological or behavioural factors over the life-course of individuals” is specifically referring to. Please revise. Following this and other suggestions made by the reviewers, we have substantially revised the Background section of the manuscript. We have now revised this specific sentence, as suggested.

□ Page 7, line 10: Should this read UK government? Please be specific. Thanks for this comment. The sentence now specifies that the Plan for Action against Childhood Obesity was published by the UK Government.

1. One thing that is missing from the background is what other similar reviews have found such as the review conducted by Khambalia et al. (2012) and how the proposed review is different and therefore needed.

Thanks for this comment. We have addressed this issue in the revised version of the Background, where we have explained why we think there is scope for a scoping review and in what ways ours will add to the body of literature already available.

• Methods

- Page 8, lines 8 – 25: Include a sub-heading under “Methods” for this section. Perhaps “Protocol” or “Protocol Design”?

We have added a subheading Protocol design under which we introduced the Arksey and O’Malley framework and its phases. We hope this will improve the clarity of this section.

- Page 8, lines 47 – 57 and page 9, lines 3 – 21: Another question that you should consider is what obesity measure is being used as a measure of effectiveness (BMI, BMI-SDS, waist circumference, waist-to-height ratio, etc.)

Thanks for this comment. We agree that it would be valuable to map out how obesity is measured and how the effectiveness of interventions is assessed. We have added a question (question 5) which focuses on the measures for obesity used in the reviews of the literature that will be included in the scoping review.

- Page 10, line 33: Should “obes” be “obese” or “obesity”?

Thanks for this comment. We use “obes*” as the truncation character * allows us to broaden our search by retrieving varying endings (e.g. obesity, obese) of our search term (“obes*”).

- Pages 12 – 13: Consider length of intervention as a main category or sub-category.

Thanks for this suggestion. We agree that the length is an important dimension of interventions to prevent obesity, alongside the intensity of the interventions delivered. Therefore, we have now added the new sub-category “Length and intensity of the intervention” under the main category “11. Description of the intervention(s)”.

• Conclusion

- Rather than a conclusion, there should be an “Ethics and Dissemination” section per author instructions.

We have revised the final section of the manuscript, as per author instructions. We have clarified that since the scoping review methodology consists of synthesizing information from publicly available publications, this study does not require ethics approval. Moreover, we give details of how we intend to disseminate the results of the scoping review once completed.

1. Whitaker RC, Wright JA, Pepe MS, Seidel KD, Dietz WH. Predicting Obesity in Young Adulthood from Childhood and Parental Obesity. *N Engl J Med.* 1997 Sep 25;337(13):869–73.
2. Guo SS, Chumlea WC. Tracking of body mass index in children in relation to overweight in adulthood. *Am J Clin Nutr.* 1999 Jul;70(1):145S–8S.
3. Juonala M, Magnussen CG, Berenson GS, Venn A, Burns TL, Sabin MA, et al. Childhood Adiposity, Adult Adiposity, and Cardiovascular Risk Factors. *N Engl J Med.* 2011;365(20):1876–85.
4. Deshmukh-Taskar P, Nicklas TA, Morales M, Yang S-J, Zakeri I, Berenson GS. Tracking of overweight status from childhood to young adulthood: the Bogalusa Heart Study. *Eur J Clin Nutr.* 2006 Jan 31;60(1):48–57.
5. Engeland A, Bjørge T, Tverdal A, Sjøgaard AJ. Obesity in Adolescence and Adulthood and the Risk of Adult Mortality. *Epidemiology.* 2004 Jan;15(1):79–85.
6. Kelsey MM, Zaepfel A, Bjornstad P, Nadeau KJ. Age-Related Consequences of Childhood Obesity. *Gerontology.* 2014;60(3):222–8.
7. Simmonds M, Llewellyn A, Owen CG, Woolacott N. Predicting adult obesity from childhood obesity: a systematic review and meta-analysis. *Obes Rev.* 2016 Feb;17(2):95–107.
8. WHO Commission on Ending Childhood Obesity. *Ending Childhood Obesity.* Geneva; 2016.
9. Mays N, Roberts E, Popay J. Synthesizing research evidence. In: Fulop N, Allen P, Clarke A, Black N, editors. *Studying the Organisation and Delivery of Health Services: Research methods.* London, UK: Routledge; 2001. p. 188–219.

10. Arksey H, O'Malley L. Scoping studies: towards a methodological framework. *Int J Soc Res Methodol.* 2005 Feb;8(1):19–32.
11. Burrows T, Golley RK, Khambalia A, McNaughton SA, Magarey A, Rosenkranz RR, et al. The quality of dietary intake methodology and reporting in child and adolescent obesity intervention trials: a systematic review. *Obes Rev.* 2012 Dec;13(12):1125–38.
12. Khambalia AZ, Dickinson S, Hardy LL, Gill T, Baur LA. A synthesis of existing systematic reviews and meta-analyses of school-based behavioural interventions for controlling and preventing obesity. *Obes Rev.* 2012 Mar;13(3):214–33.
13. Mallonee LF, Boyd LD, Stegeman C. A scoping review of skills and tools oral health professionals need to engage children and parents in dietary changes to prevent childhood obesity and consumption of sugar-sweetened beverages. *J Public Health Dent.* 2017 Jun;77 Suppl 1:S128–35.
14. Rivera J, McPherson A, Hamilton J, Birken C, Coons M, Iyer S, et al. Mobile Apps for Weight Management: A Scoping Review. *JMIR mHealth uHealth.* 2016 Jul 26;4(3):e87.
15. Waters E, de Silva-Sanigorski A, Burford BJ, Brown T, Campbell KJ, Gao Y, et al. Interventions for preventing obesity in children. In: Waters E, editor. *Cochrane Database of Systematic Reviews.* Chichester, UK: John Wiley & Sons, Ltd; 2011.
16. de Melo Boff R, Araujo Liboni RP, de Azvedo Batista IP, Heineck de Souza L, da Silva Oliveira M. Weight loss interventions for overweight and obese adolescents: a systematic review. *Eat Weight Disord.* 2017;22:211–29.
17. Al-Khudairy L, Loveman E, Colquitt JL, Mead E, Johnson RE, Fraser H, et al. Diet, physical activity and behavioural interventions for the treatment of overweight or obese adolescents aged 12 to 17 years. In: Rees K, editor. *Cochrane Database of Systematic Reviews.* Chichester, UK: John Wiley & Sons, Ltd; 2017.
18. Mead E, Brown T, Rees K, Azevedo LB, Whittaker V, Jones D, et al. Diet, physical activity and behavioural interventions for the treatment of overweight or obese children from the age of 6 to 11 years. ELLS LJ, editor. Vol. 2017, *Cochrane Database of Systematic Reviews.* Chichester, UK: John Wiley & Sons, Ltd; 2017.
19. Barshop NJ, Francis CS, Schwimmer JB, Lavine JE. Nonalcoholic fatty liver disease as a comorbidity of childhood obesity.
20. Félix DR, Costenaro F, Bertaso C, Gottschall A, Perdomo Coral G. Non-alcoholic fatty liver disease (NAFLD) in obese children-effect of refined carbohydrates in diet.
21. Cohen AK, Rai M, Rehkopf DH, Abrams B. Educational attainment and obesity: a systematic review. *Obes Rev.* 2013 Dec;14(12):989–1005.
22. Miller AL, Lee HJ, Lumeng JC. Obesity-Associated Biomarkers and Executive Function in Children. *Pediatr Res.* 2014 Oct 13;77(1–2):143–7.
23. Pizzi MA, Vroman K. Childhood Obesity: Effects on Children's Participation, Mental Health, and Psychosocial Development. *Occup Ther Heal Care.* 2013 Apr 19;27(2):99–112.
24. Buttitta M, Iliescu C, Rousseau A, Guerrien A. Quality of life in overweight and obese children and adolescents: a literature review. *Qual Life Res.* 2014 May 19;23(4):1117–39.
25. Tsiros MD, Olds T, Buckley JD, Grimshaw P, Brennan L, Walkley J, et al. Health-related quality of life in obese children and adolescents. *Int J Obes.* 2009 Apr 3;33(4):387–400.
26. Biro FM, Wien M. Childhood obesity and adult morbidities. *Am J Clin Nutr.* 2010 May;91(5):1499S–1505S.
27. Hill JO, Wyatt HR, Peters JC. Energy balance and obesity. *Circulation.* 2012 Jul 3;126(1):126–32.
28. Spruijt-Metz D. Etiology, Treatment, and Prevention of Obesity in Childhood and Adolescence: A Decade in Review. *J Res Adolesc.* 2011 Mar;21(1):129–52.

VERSION 2 – REVIEW

REVIEWER	Elise C. Brown
-----------------	----------------

	Oakland University, USA
REVIEW RETURNED	14-Jan-2018

GENERAL COMMENTS	Most of my previous comments were addressed by the author, however, there are a few more points that need to be addressed. Abstract Page 2, lines 21-23: Please correct grammar when introducing the research question. Methods Page 4, line 35: Please replace the word “nowadays” with a more formal term. Page 4, line 37: It seems like a word is missing from the text: “hence cannot afforded at scale.” For the “Supplementary material 2,” it currently reads “PRIMA” rather than “PRISMA.”
---